# Optimized Process for Melt Pyrolysis of Methane to Produce Hydrogen and Carbon Black over Ni Foam/NaCl-KCl Catalyst

**Mengying Liu** [1], **Zeai Huang** [1,2,*], **Yunxiao Zhou** [1], **Junjie Zhan** [1], **Kuikui Zhang** [1], **Mingkai Yang** [1], **Ying Zhou** [1,2,3,*]

[1] School of New Energy and Materials, Southwest Petroleum University, Chengdu 610500, China
[2] State Key Laboratory of Oil and Gas Reservoir Geology and Exploitation, Southwest Petroleum University, Chengdu 610500, China
[3] Tianfu Yongxing Laboratory, Chengdu 610217, China
* Correspondence: zeai.huang@swpu.edu.cn (Z.H.); yzhou@swpu.edu.cn (Y.Z.)

**Abstract:** Methane pyrolysis transforming $CH_4$ into hydrogen without a $CO_2$ byproduct is a potential hydrogen production process under the net-zero emission target. The melt pyrolysis of methane is a technology that could simultaneously obtain hydrogen and carbon products. However, its catalytic activity and stability are still far from satisfactory. In this work, a new strategy for the melt pyrolysis of methane to hydrogen production was proposed using Ni foam and molten NaCl-KCl. The increase in the amount of Ni foam was found to enhance the methane conversion rate from 12.6% to 18%. The process was optimized by the different amounts of catalysts, the height of the Ni foam layer, and the filling method of Ni foam, indicating that the methane conversion rate of the string method could reach 19.2% at 900 °C with the designed aeration device. Furthermore, we observed that the addition of molten salt significantly alleviated the carbon deposition deactivation of the Ni foam and maintained its macrostructure during the reaction. The analysis of the carbon products revealed that carbon black could be obtained.

**Keywords:** melt pyrolysis of methane; Ni foam; carbon black; hydrogen

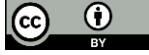

## 1. Introduction

With the rapid development of the economy comes the exposure of human society to heavy environmental pollution due to the large consumption of fossil energy with greenhouse gas emissions. A global consensus has been reached to build a clean and sustainable society under a net-zero emission target. As a result, hydrogen energy has received widespread attention because of its advantages such as zero carbon emissions with a high combustion calorific value, high energy density, and easy storage [1]. As a clean fuel and energy carrier, hydrogen is of great significance in reducing greenhouse gas emissions [2] and improving energy supply security and economic competitiveness. However, 96% of hydrogen is produced from fossil fuels with a large number of $CO_2$ emissions [3]. Common hydrogen production methods from methane include the steam reforming of methane, partial oxidation of methane, dry reforming of methane, and methane pyrolysis [4–6]. In total, 48% of hydrogen comes from methane steam reforming (Equation (1)) to produce hydrogen without $CO_2$ capture and storage named gray hydrogen [7,8]. Dry reforming of methane can reduce $CO_2$ emissions, but the high cost of equipment and the purification of hydrogen hinder its industrial application. Therefore, it is highly demanded to change the method of hydrogen production and obtain hydrogen production without $CO_2$ emissions.

Recently, the methane pyrolysis (Equation (2)) [9] reaction has gained a lot of attention as a new technology for environmentally friendly hydrogen production.

$$CH_4 + 2H_2O \rightarrow CO_2 + 4H_2 \qquad \Delta H^0_{298K} = 252.75 \text{ kJ/mol} \tag{1}$$

$$CH_4 \rightarrow 2H_2 + C \qquad \Delta H^0_{298K} = 74.85 \text{ kJ/mol} \tag{2}$$

Within these processes, methane pyrolysis does not directly produce $CO_x$ [10]. Compared with steam methane reforming, it shows more application prospects with regard to the net-zero emission target [11,12]. In addition, the carbon products such as carbon nanotubes and graphene [13–15] obtained from methane pyrolysis create additional economic value. This technology was developed in the 20th century and can be mainly divided into non-catalytic pyrolysis, plasma thermal decomposition, catalytic pyrolysis, and molten medium pyrolysis [16]. Due to the high activation energy required by the C-H bond [17], the non-catalytic pyrolysis temperature needs to be higher than 1200 °C [16,18], requiring large energy consumption. The catalytic methane pyrolysis developed can significantly reduce the activation energy of methane cracking to hydrogen with a reduced temperature of 600–900 °C [19,20]. Although some catalysts show excellent catalytic activity, such as Ni, Fe, and Co, they easily and rapidly deactivate due to the rapid agglomeration of metal particles and carbon deposition. The separation of carbon products and catalysts makes the process more complicated and with economically high costs, hindering the continuity and scalability of the process. To solve the problem of catalyst deactivation during conventional methane pyrolysis, Daniel Tyrer [21] proposed the concept of the melt pyrolysis of methane (MPM) in the early 20th century. The problems of catalyst deactivation and the difficult separation of carbon products in traditional pyrolysis processes were solved by methane pyrolysis with a molten medium. The carbon products during the reaction float on the surface of the melt, effectively preventing the catalyst from being inactivated by carbon deposition [22]. More importantly, such a process does not directly produce $CO_x$; this means that methane can be transformed into hydrogen from gray hydrogen to blue hydrogen via melt pyrolysis.

Globally, the hydrogen production technology from melt pyrolysis methane has not yet achieved commercial application, mainly due to its high energy consumption. Developing efficient catalysts and reaction systems to achieve the efficient melt pyrolysis of methane is the key to reducing reaction energy consumption. Catalysts for the melt pyrolysis of methane can be divided into four categories, including single metal, alloy, molten salt, and metal/salt. The catalytic effects of Bi, In, Sn, Ga, Ni, Pt, Pd, and other molten metals on methane pyrolysis were investigated using the methane conversion rate as the index. These metals were studied as active metal catalysts (Ni, Pd, and Pt) and low melting point solvent metals (In, Sn, Ga, and Bi) [23–25]. It was found that an alloy of $Ni_{0.17}Bi_{0.83}$ offered the highest rate of hydrogen production ($9 \times 10^{-8}$ mol $H_2/cm^2 \cdot s$), and $Ni_{0.17}Bi_{0.83}$ achieved 95% in a 1.1 m column bubble reactor at 1065 °C. However, single-metal catalysis usually shows low conversion efficiency and high-temperature requirements. The activity under similar conditions could be improved over alloys [26–28] composed of active metals and low melting point metals. However, whether it is single-metal catalysts or alloy catalysts, the content of metal impurities in the carbon products is too high and difficult to remove. Recent research has proposed the use of salt to replace metal [29,30], that is, molten salt catalytic methane pyrolysis, because salt can be readily removed from the carbon products by simple processes such as washing in water and heating. Parkinson et al. [29] studied methane pyrolysis in different molten salts: NaBr, KBr, NaCl, KCl, and NaBr/KBr (48.7/51.3 mol%). They found that the conversion of NaCl and KCl achieved 5.46% and 5.23% in a quartz reactor at 1000 °C. Compared with a conventional metal catalyst and molten metals, its methane conversion rate is limited [31] due to the absence of an active metal for methane cracking. Considering the high methane conversion rate of molten metal and the low impurity rate of molten salt, some researchers combined the two processes [32], aiming to obtain a high methane conversion and low impurity content of carbon products at the same time. The mix of molten metal and molten salt as a catalyst also contributes to purifying the carbon from residual metal contamination. Nazanin Rahimi

et al. [33] experimentally studied molten metal (Ni-Bi) with molten salts (NaBr) in methane pyrolysis and reported a conversion of less than 15% at 920 °C but that the methane conversion could reach 38% when the temperature was increased to 1000 °C. However, compared with alloy catalysis the catalytic activity is still unsatisfactory. The main reason is that the dissolution of active metals in molten salts is still limited and the diffusion of methane in the solid and liquid phases was much slower than that in the gas phase. Designing catalysts and processes to improve methane diffusion and activation is challenging but of great urgency.

Herein, we proposed a new configuration of melt pyrolysis of methane for improved hydrogen production efficiency and carbon black formation. Nickel catalysts are known for their higher catalytic activities but they quickly deactivate above 600 °C because the carbon byproduct encapsulates its active sites. In addition, methane pyrolysis in molten media has proven to be advantageous over conventional pyrolysis. The introduction of the liquid reaction interface can efficiently eliminate the risk of interfacial carbon deposition. Catalytically, alloys usually have a melting point higher than 1000 °C, however, the molten salt system can reduce the melting point by adjusting the ratio and components of the molten salt. The salts used are also cheaper than metals, which should enhance the economic feasibility of methane cracking and process scale-up. Therefore, we dispersed Ni foam in molten salts, expecting to bring a high methane conversion at a lower temperature with the system resisting deactivation. We specifically addressed the following issues related to $CH_4$ pyrolysis in Ni foam/NaCl-KCl: (1) Does the molten salt in the system alleviate metal catalyst deactivation? (2) Is the Ni foam/NaCl-KCl activity stable during sustained operation? (3) Does the aeration device have an effect on methane conversion? The process was optimized by using different amounts of catalysts, varying the height of the Ni foam layer and the filling method of Ni foam, and trying different aeration devices. The effects of these conditions on the performance of the melt pyrolysis of methane were then studied. Using Ni foam and molten NaCl-KCl as the catalysts, we succeeded in improving the methane conversion rate with the designed aeration device. It is expected to obtain a higher methane conversion and more pure carbon products.

## 2. Experimental Methods

### 2.1. Materials

Commercial Ni foam (Kunshan Tengerhui Electronic Technology Co., Ltd., Suzhou, China) was used as the metal catalyst. It was cut into small 1 × 1 cm pieces with a thickness of 1 cm. An aperture of 20 ppi and NaCl and KCl (KESHI) was used to prepare the molten salt catalysts.

### 2.2. Melt Pyrolysis of Methane Process

A schematic diagram of the experimental setup is shown in Figure 1. In this diagram, four different sections were considered.

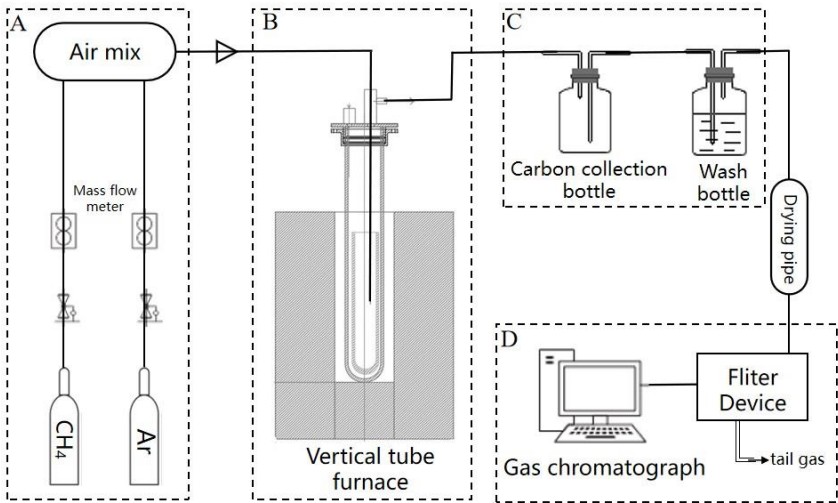

**Figure 1.** Diagram of the methane pyrolysis process: (**A**) air mix section; (**B**) reaction section; (**C**) carbon product collection section; and (**D**) reaction gas section.

Section A is the air mix section that flows to the reactor system; mass flow controllers (MFC) were used to control the $CH_4$ and Ar gas.

Section B is the reaction section. First, the gas tube was inserted from the top and Ni foam was weighed and evenly dispersed in the bottom of a corundum reaction tube (inner diameter 27 mm; hemispherical bottom), then NaCl and KCl with equal molar ratios were added, and it was ensured that the salts were covered with the Ni foam. The portion of the corundum tube reactor filled with the catalyst was mounted in a temperature-controlled, electrically-heated furnace. For pyrolysis, the reactant gas was administered through a small diameter corundum tube (the diameter of the tube is 4 mm) that was inserted from the top. The temperature was measured using K-type thermocouples.

Section C is the carbon product collection section. A carbon collection bottle was used to collect the carbon. During the reaction, some carbons were blown by the gas into the carbon collection bottle. A wash bottle was used to prevent gas backflow and back suction and a drying pipe was used to separate the water in the reacted gas.

Section D is the reaction gas section. A filter device was used to filter out impurities in the reaction gas. The gaseous products were determined by a gas chromatograph (GC-7820A, C.Name: TDX-01, Type: Φ3 mm × 1.5 mm). A thermal conductivity detector (TCD) was used to monitor the $H_2$ consumption and a flame ionization detector (FID) was used to monitor the $CH_4$ consumption.

In the experiment, the temperature was elevated from room temperature to 900 °C at a rate of 10 °C/min under Ar flow. Initially, only Ar gas (100 sccm) was fed into the reaction; Ar gas was used as a purge gas to create an oxygen-free environment for the reaction until the system reached the operating temperature, then it was kept for 420 min at 900 °C. After the temperature was reached and held, the $CH_4$ gas was flowed (99.9% 100 sccm $CH_4$) into the tube and the exit gas from the reactor went through the automatic sampling loop of a gas chromatograph every 30 min. During the reaction process, the carbon products floated on the molten salt, and part of the carbon fell into the external corundum tube and blew into the carbon collection bottle. After the reaction, the corundum reaction tube was cooled and removed to collect the carbon to generate carbon products. The collected carbon was transferred into a beaker and washed with deionized water. The carbon products were then filtered, rinsed, and rewashed. The calculation for the $CH_4$ conversion is:

$$\chi_{CH_4} = \frac{[CH_4]_{out} - [CH_4]_{in}}{[CH_4]_{in}} \tag{3}$$

where $[CH_4]_{out}$ and $[CH_4]_{in}$ are the mass flow rate of $CH_4$ into and out of the reactor.

*2.3. Characterizations*

The morphology of the recovered carbon sample was analyzed by various analytical methods. Scanning electron images were obtained at 10 kV using a scanning electron microscope (SEM) by Thermo scientific which provided high-resolution images. X-ray diffraction showed peaks of the various components of the sample and measurements were performed using an X'Pert PRO (PAN-alytical, Malvern, UK) with Cu-K$\alpha$ radiation ($\lambda$ = 1.541 nm). The diffraction pattern was collected for $2\theta$ = 5–85° with a step size of 0.02°. Raman spectroscopy was employed to study the crystallinity of the carbon sample. Raman spectra of the carbon from a 532 nm excitation were obtained using a Horiba Jobin Yvon LabRAM Aramis spectrometer (Horiba Jobin Yvon, Palaiseau, France).

## 3. Results and Discussion

### 3.1. Effect of Ni Foam/NaCl-KCl Ratio on Methane Conversion

Firstly, the blank test without salts showed much higher catalytic conversion activity at the beginning of the reaction (Figure S1), and the 2 g Ni foam and the methane conversion reached 50% at 900 °C. However, the conversion decreased to 28% after a 12 h reaction due to the heavy carbon deposition. More importantly, we took out the catalyst in the tube after the reaction, and the macrostructure of the Ni foam without salt had been burned and dispersed, resulting in small particle black products (Figure S2), indicating that the reuse of the catalyst was impossible. In the case of Ni foam/NaCl-KCl, the Ni foam around the inlet gas pipe showed carbon deposition, while the other areas showed no obvious change, indicating that the molten salt alleviated the metal catalyst deactivation and that the Ni foam was almost unaffected by carbon deposition, and the methane conversion decreased only slightly. The results showed that the addition of molten salt significantly alleviated the carbon deposition deactivation of Ni foam and maintained its macrostructure. The reason for the stability of the molten salt-promoted Ni foam/NaCl-KCl catalyst is that molten salt has better wettability for Ni foam, and a liquid film of molten salt may be formed on the surface of the Ni foam, thereby preventing the deposition of carbon leading to the encapsulation of the catalyst. However, the conversion rate decreased compared with the group without molten salt [34,35]. This decrease in conversion rate may be due to the molten salt liquid environment obstructing the diffusion of $CH_4$ gas, which greatly reduces the contact surface between the $CH_4$ gas and Ni foam, resulting in a much lower conversion than a gas-solid reaction. Therefore, the effect of the Ni foam/NaCl-KCl ratio on methane conversion was studied. We gradually increased the amount of Ni foam, using 0, 2, 10 g, and 20 g of Ni foam with a molten salt constant of 150 g for the melt pyrolysis of methane. The results (Figure 2) showed a methane conversion rate of over 10 g of Ni foam was up to 18% at 900 °C while the methane conversion rate of 2 g and 20 g could reach 14% and 16.5%, respectively. The conversion rate of methane was only around 12.6% without the active metal of Ni foam. This indicated that the addition of Ni foam into the salt of NaCl/KCl doubled the methane conversion.

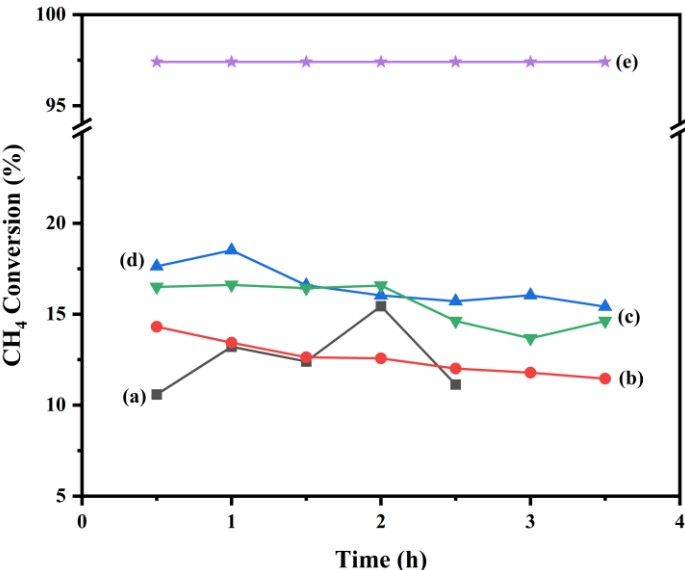

**Figure 2.** CH$_4$ Conversion of catalysts: (a) NaCl-KCl; (b) 2 g Ni/NaCl-KCl; (c) 20 g Ni/NaCl-KCl; (d) 10 g Ni/NaCl-KCl; and (e) equilibrium conversion. (Conditions: 99.9% 100 sccm CH$_4$; 900 °C; 20 ppi Ni foam; 150 g NaCl-KCl).

After increasing the amount of Ni foam in the system, it was found that the conversion rate was not significantly improved, which might be due to the Ni foam being dispersed in the molten salt without sufficient contact with the CH$_4$ gas. In order to investigate the contact state between the reaction gas and catalyst, we used quartz tubes and water to simulate ventilation experiments. We found that in the liquid environment, the gas raised around the intake pipe and only touched the Ni foam which was close to the intake pipe, instead of dispersing into the Ni foam with small bubbles. Moreover, the amount of Ni foam determined the height of the nickel layer. If the amount of Ni foam was increased, the height of the nickel layer would increase, and the contact time between the reaction gas and the nickel layer could be slightly prolonged. In addition, the position of the Ni foam in the tube may also affect the contact with the reaction gas. Therefore, we considered using an aerator to disperse the CH$_4$ gas to enhance the dispersion of bubbles in the molten salt during the reaction, which might increase the contact area with Ni foam.

### 3.2. Effect of Aeration Device on Methane Conversion

We studied the effect of aeration devices on methane conversion by customizing different types of aeration devices and simulating them in water. We were trying to understand if increasing the height of the Ni foam layer, changing the filling method of the Ni foam, and adding different aeration devices could increase the gas-solid contact area and improve the methane conversion. Experiments were carried out on the premise of keeping the total intake velocity and reaction temperature unchanged (99.9% 100 sccm CH$_4$; 900 °C). Table 1 lists the different Ni foam filling methods and different aeration devices used (Figure 3). Figure 3a is the normal case that we studied to perform the reaction, i.e., the Ni foam was fixed on the gas tube to prevent it from floating with the gas, gas bubbles rose around the Ni foam, and the contact area between the gas and Ni foam was increased. Figure 3b shows quartz sand aeration where the corundum tube was placed into 2 cm deep quartz sand. This method solved the problem of bubble dispersion and the bubbles were smaller and denser than without quartz sand. Figure 3c indicates a Ni mesh-wrapped gas pipe with iron wire where the Ni mesh had a diameter of 10 cm. Figure 3d shows a four-hole tube with hole diameters of 1 mm. After simulated gas was injected into water, it was found that different aeration devices showed certain dispersion effects on the bubbles at high flow rates [36,37]. The four-hole tube and Ni mesh increased the number of bubbles. In aerator head experiments, the bubbles mainly emerged from the

gap between the quartz aerator head and corundum tube and did not appear to play a significant role in the dispersion. Therefore, we used an integrated quartz aerator, where the diameter of the gas tube is 4 mm and the diameter of the aerator head is 2 cm, with a gas hole of 1.0 mm, as shown in Figure 3e, but due to the pressure and gas flow rate, the gas mainly emerged from the upper hole, and there were no obvious bubbles from the side face.

**Table 1.** Experimental conditions of gas dispersion with different aeration devices [a].

| Ni Foam Shape | Ni Foam Size | Gas Dispersion Methods |
|---|---|---|
| Cube | 1 × 1 × 1 cm | Φ4 mm gas pipe |
| Fragmental | - | Φ4 mm gas pipe |
| Circular (String) | Φ2 cm, 1 cm | Φ4 mm gas pipe |
| Cube | 1 × 1 × 1 cm | Φ1 mm four-hole gas pipe |
| Cube | 1 × 1 × 1 cm | Disperser device, Φ1 mm |
| Cube | 1 × 1 × 1 cm | Integrated disperser device, Φ1 mm |
| Cube | 1 × 1 × 1 cm | Ni mesh cover gas pipe |

[a] Conditions: 99.9% 100 sccm $CH_4$; 900 °C; 10 g Ni foam (Ni content 20%); 150 g NaCl-KCl.

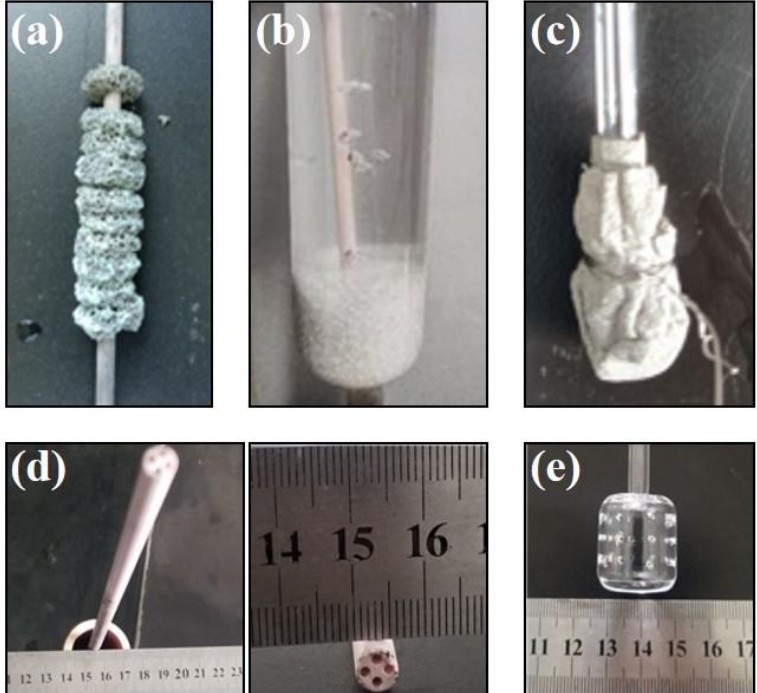

**Figure 3.** (**a**) Ni foam string; (**b**) quartz sand; (**c**) nickel mesh; (**d**) four-hole gas pipe; and (**e**) integrated disperser device.

The reaction results of the methane conversion using these different aeration devices with 10 g Ni foam and 150 g NaCl-KCl is shown in Figure 4. It could be seen that the above aeration devices showed obviously different methane conversion efficiency. Each aeration device showed significant improvement in methane conversion compared with the pure salt group (Figure 4A), and curve (a) showed the best activity in the experiment. The conversion rate of Ni mesh reached 18.9% at 900 °C shown in Figure 4B. Quartz sand and the four-hole tube aeration slightly dispersed the gas which improved the conversion rate compared with the normal case of Ni/NaCl-KCl. However, such conversion was not improved, and the stability was not as good as compared with the case of the Ni foam string. We speculated that it might be because when the mesh was fixed on the gas pipe, it was

not easily swept by the airflow, and the contact time with the gas increased. In the Ni foam string experiment, the Ni foam (total:16; diameter of 2 cm and height of 1 cm) was fixed on the gas tube. After a 5 h reaction, the conversion rate increased by more than 19% at 900 °C and showed good stability. From these results, we could conclude that the aeration device affected the conversion rate for the melt pyrolysis of methane with the optimized device.

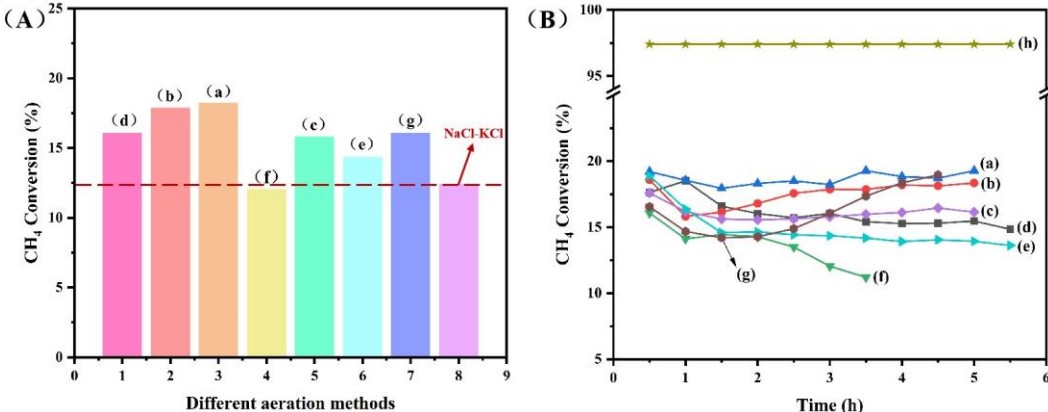

**Figure 4.** (**A**) CH₄ conversion and (**B**) CH₄ conversion under different aeration devices: (a) Ni (string)/NaCl-KCl; (b) Ni (fragment)/NaCl-KCl; (c) four-hole tube; (d) Ni/NaCl-KCl; (e) integrated disperse device; (f) quartz sand; (g) Ni mesh; and (h) equilibrium conversion. (Conditions: 99.9% 100 sccm CH₄; 900 °C; 10 g Ni foam; 20 ppi Ni foam; 150 g NaCl-KCl).

### 3.3. Analysis of Carbon Product

The solid carbon product was extracted after the completion of the reaction when the reactor cooled down to room temperature. The morphology of the solid carbon mixture produced after the melt pyrolysis of the methane reaction was analyzed using SEM. Figure 5a,b shows the morphology of the carbon product in the Ni foam/NaCl-KCl system which showed the best catalytic activity. The carbon particles exhibited a nanocrystalline structure and assembled into larger, spherical cauliflower-like carbon structures with no large-scale ordering or graphitic nature. These micron-scale shapes were consistent with amorphous carbon structures such as carbon black which are typically synthesized in the gas-phase decomposition of methane [31]. Figure 5c represents the XRD data from the carbon obtained from CH₄ pyrolysis with the Ni foam/NaCl-KCl. The XRD patterns of the carbon products show diffraction peaks at 25° assigned to the presence of carbon black [38,39]. Furthermore, the peak of carbon black is shifted at lower angles, and the peak shift towards lower angles was reported in the literature and is called salt ion intercalation, which is explained as an expansion of the structural disorder within the carbon layers by pushing some atoms out of the plane [29]. NaCl and KCl peaks were also shown in the figure, which may be due to the residual salt. In addition, some salt particles were observed on the carbon product (Figure 5b), which also corresponds to the results of salt peaks in XRD. The solid carbon was completely intermixed with the molten salt and solidified salt after cooling and could not be fully separated by the water wash.

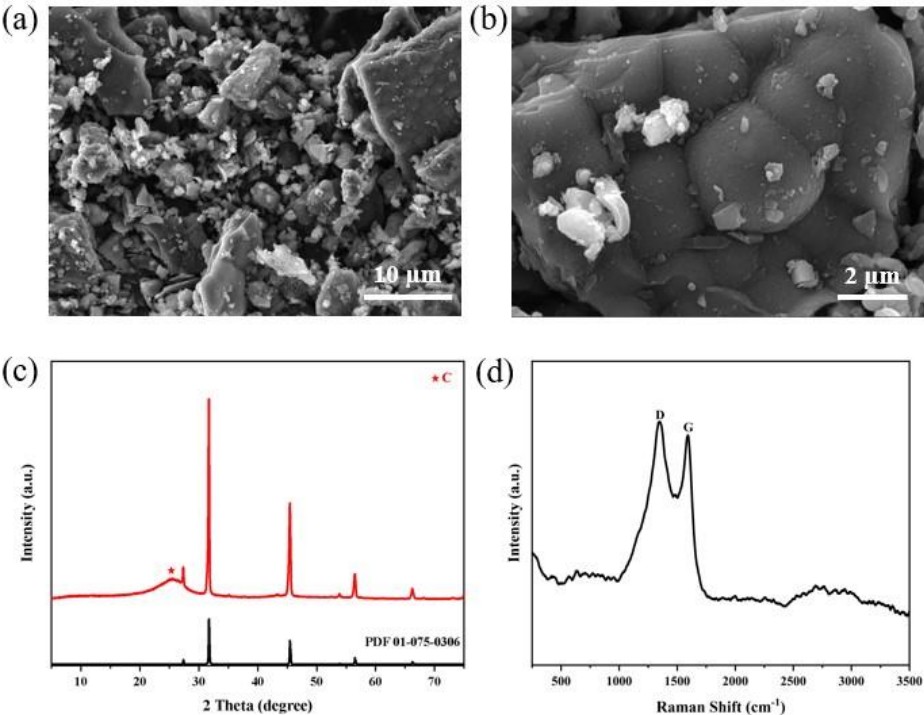

**Figure 5.** (**a**,**b**) SEM images of the carbon product; (**c**) X-ray diffraction pattern of the carbon product; and (**d**) Raman spectra of the carbon product.

To better understand the short-range structure of the formed carbon materials, Raman spectroscopy was performed, with the results shown in Figure 5d. The ratio of the peak intensities $I_D/I_G$ indicates the degree of graphitization. When the $I_D/I_G$ ratio decreases, the degree of graphitization increases; a larger ratio indicates an amorphous structure. The Raman spectra present three obvious characteristic bands which are D, G, and 2D bands. The D band ("disorder" band) centered at 1350 cm$^{-1}$ is derived from the characteristic vibrations of edge and/or defected carbon sites. The G band ("graphite" band) centered at 1590 cm$^{-1}$ is associated with the stretching frequency of C-C bonds in the graphitic basal plane [40,41]. The observed intensity ratio of the D to G bands ($I_D/I_G$) of the carbon product from Ni foam/NaCl-KCl is approximately 1.05, consistent with an amorphous structure, and these features are akin to a carbon black structure in which the degree of graphitization is extremely low [42]. In the Raman spectra of amorphous carbons such as carbon black, the convolution of the D and G bands is severe [31]. From the results of the SEM, XRD, and Raman spectra, it can be concluded that more amorphous carbon was generated in the Ni foam/NaCl-KCl system.

## 4. Conclusions

In this paper, commercial Ni foam combined with molten salt was used in methane pyrolysis to produce hydrogen and carbon black. It is concluded that the Ni foam without the molten salt has a relatively short active life and gradually deactivates after 4 h of operation. This occurs because the macrostructure of the Ni foam has been burned and the reuse of the catalyst is impossible. On the contrary, the Ni foam catalyst with molten salt has a long-term stability of about 12 h without any deactivation. This indicates that the molten salt could significantly relieve the carbon deposition on Ni foam. The conversion rate of methane was improved by increasing the amount of Ni foam. The optimized 10 g Ni foam and 150 g NaCl-KCl showed good activity at 900 °C. The results clearly indicated that Ni foam in molten salts shows better methane conversion at a lower temperature. In

addition, the use of aeration devices showed obvious differences in the methane conversion efficiency, and the methane conversion rate of the string method could reach 19.2% at 900 °C with the designed aeration device, simultaneously producing a separable solid carbon product due to the low wettability of the carbon produced by the melt. In conclusion, this work could provide a new process for the conversion of methane to hydrogen and carbon black.

**Supplementary Materials:** The following supporting information can be downloaded at: https://www.mdpi.com/article/10.3390/pr11020360/s1, Figure S1: Comparison of catalyst stability (Conditions: 99% 100 sccm $CH_4$; 900 °C; 20 ppi Ni Foam; 150 g NaCl-KCl); Figure S2: Comparison of Ni Foam before and after: (a) Before reaction, (b) After reaction.

**Author Contributions:** Conceptualization, Z.H. and Y.Z. (Ying Zhou); Methodology, M.L., Y.Z. (Yunxiao Zhou), J.Z., and K.Z.; Formal analysis, Y.Z. (Yunxiao Zhou), J.Z., K.Z., M.Y., and Y.Z. (Ying Zhou); Investigation, Y.Z. (Yunxiao Zhou); Data curation, M.L.; Writing—original draft, M.L.; Writing—review and editing, Z.H.; Supervision, Z.H. and Y.Z. (Ying Zhou); Project administration, Y.Z. (Ying Zhou) All authors have read and agreed to the published version of the manuscript.

**Funding:** This research was funded by the Special project for the central government to guide the development of local science and technology in Sichuan Province (No. 2021ZYD0099), the National Natural Science Foundation of China (No. 202209136), the Sichuan Provincial International Cooperation Project (No. 2021YFH0055 and 2022YFH0084), and the Sichuan Provincial Key Research and Development Project (No. 22ZDYF3690).

**Data Availability Statement:** Not applicable.

**Conflicts of Interest:** The authors declare that they have no conflict of interest.

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
