# Peer review of "Optimized Process for Melt Pyrolysis of Methane to Produce Hydrogen and Carbon Black over Ni Foam/NaCl-KCl Catalyst"

_processes, doi:10.3390/pr11020360_

Round 1
Reviewer 1 Report
Although this is a very meaningful research work, it is obviously too simple. There was no detailed study. Therefore, it is recommended to reject the paper.
Author Response
We are grateful to the reviewer for the positive and constructive evaluation of our work that “this is a very meaningful research work”. In this work we dispersed Ni foam in molten salts, expecting to bring high methane conversion at lower temperature with system resisting deactivation, which was not reported before. With the designed aeration devices and Ni foam catalysts, we found the amount of Ni foam was increased, the height of the nickel layer would increase, and the contact time between the reaction gas and the nickel layer could be slightly prolonged with the formation of carbon black. Therefore, we use different aerator to disperse the CH4 gas to enhance the dispersion of bubbles in molten salt during reaction, which increase the contact area with Ni foam. This work provided a new process for the conversion of methane to hydrogen and carbon black.
Reviewer 2 Report
The manuscript from M. Liu and co-workers reported the optimized process for melt pyrolysis of methane to produce hydrogen and carbon black over Ni foam/NaCl-KCl catalyst. The several processes have been investigated. The paper is well organized and concise in its analysis. This article can be published after a major revision. Here are my comments/suggestions in order to improve the present form:
1. There are some editorial mistakes that should be revised.
2. The novelty of this paper is questionable. The degree of novelty is not presented. What is unique or different from the existing literature? Please also mention the existing literature related to the catalytic study of the NaCl-KCl catalyst for melt pyrolysis of methane in the introduction.
3. Please revise the Figure S1, the break in x-axis (at 5h) is not necessary. Please plot the full graph.
Author Response
The manuscript from M. Liu and co-workers reported the optimized process for melt pyrolysis of methane to produce hydrogen and carbon black over Ni foam/NaCl-KCl catalyst. The several processes have been investigated. The paper is well organized and concise in its analysis. This article can be published after a major revision. Here are my comments/suggestions in order to improve the present form:
2.1.There are some editorial mistakes that should be revised.
Response 2.1: We thank the reviewer for the valuable comments. We have revised it in the manuscript.
2.2. The novelty of this paper is questionable. The degree of novelty is not presented. What is unique or different from the existing literature? Please also mention the existing literature related to the catalytic study of the NaCl-KCl catalyst for melt pyrolysis of methane in the introduction.
Response 2.2: We thank the reviewer for the valuable comments. The conventional metal catalysts are known by their higher catalytic activities, but it quickly deactivates above 600℃, the introduction of the liquid reaction interface can efficiently eliminates the risk of interfacial carbon deposition, catalytically alloys usually have a melting point higher than 1000℃, however, the molten salt system can reduce the melting point by adjusting the ratio and components of the molten salt. Therefore, we dispersed Ni foam in molten salts, expecting to bring high methane conversion at lower temperature with system resisting deactivation. And we also added the existing literature related to the catalytic study of the NaCl-KCl catalyst for melt pyrolysis of methane in the introduction. Parkinson et al., Who reported methane pyrolysis in different molten salts: NaBr, KBr, NaCl, KCl and NaBr/KBr (48.7/51.3 mol%). The researcher found the conversion of NaCl and KCl achieved 5.46% and 5.23% (International Journal of Hydrogen Energy 46, 6225-6238 (2021)). Nazanin Rahimi et al. reported the molten metal (Ni-Bi) with molten salts (NaBr) in methane pyrolysis and reported a conversion is less than 15% at 920℃ and the methane conversion could reach 38% when increased the temperature to 1000℃ (Applied Catalysis B: Environmental 254, 659-666 (2019)). The explanatory texts were added to page 2: line 79-81 and line 86-89 (highlighted in blue).
2.3. Please revise the Figure S1, the break in x-axis (at 5h) is not necessary. Please plot the full graph.
Response 2.3: We thank the reviewer for the valuable comments. We have revised the Figure S1 in SI.
Reviewer 3 Report
Overall Evaluation and Recommendations
However, this manuscript represented the excellent results, but there are some questions that need to be addressed before the manuscript can be accepted for publication. Increase the discussion and more explain the characterizations. Accordingly, I would like to recommend it for publication after major revision.
Questions:
1) It seems that by the addition of more numerical data in the abstract section, you will able to illustrate the research novelty, more clearly. Add more numerical results to the abstract section.
2) In the literature review, please use more recent works especially the ones which were published in 2020, 2021 and 2022.
3) Moreover, in the introduction section you can use of some other papers in the filed CO2 reforming of CH4 or dry reforming of methane (DRM) or combination of partial oxidation of CH4 and DRM. Therefore, introduction will be better. For example you can use of these papers:
S. M. Sajjadi, M. Haghighi, F. Rahmani, "On the synergic effect of various anti-coke materials (Ca–K–W) and glow discharge plasma on Ni-based spinel nanocatalyst design for syngas production via hybrid CO2/O2 reforming of methane", International Journal of Energy Research, 2022, 108: p. 104810
S. M. Sajjadi, M. Haghighi, J. Eshghi, "Synergic influence of potassium loading and plasma-treatment on anti-coke property of K-promoted bimetallic NiCo-NiAl2O4 nanocatalyst applied in O2-Enhanced dry reforming of CH4", International Journal of Hydrogen Energy, 2019, 44 (26): p.13397-13414
S. M. Sajjadi, M. Haghighi, F. Rahmani, J. Eshghi, "Plasma-enhanced sol-gel fabrication of CoWNiAl2O4 nanocatalyst used in oxidative conversion of greenhouse CH4/CO2 gas mixture to H2/CO", Journal of CO2 Utilization, 2022, 61: p. 102037
4) Please add the specifications of the utilized column in gas chromatography? The name of that, is packed column or capillary?
5) CH4 conversion formula (in page 3, equation number 3), it seem that it is wrong. You must write in this form
(CH4 (out) – CH4 (in))/(CH4 (in))
6) If it is possible for you, please compare your results with the other papers. Therefore, superiority of your work will be clear.
Author Response
However, this manuscript represented the excellent results, but there are some questions that need to be addressed before the manuscript can be accepted for publication. Increase the discussion and more explain the characterizations. Accordingly, I would like to recommend it for publication after major revision.
Response 3: We are grateful to the reviewer for the positive and appreciative evaluation of our work, and for pointing out precise issues that need to be addressed in the present version. In the following, we reply to all the helpful comments in detail.
3.1. It seems that by the addition of more numerical data in the abstract section, you will able to illustrate the research novelty, more clearly. Add more numerical results to the abstract section.
Response 3.1: We thank the reviewer for the valuable comments. We have added the numerical data in the abstract section. “the methane conversion rate of the string method could reach 19.2% at 900℃ with the designed aeration device. Furthermore, we observed that the addition of molten salt significantly alleviated the carbon deposition deactivation of Ni foam and maintained its macro structure”. The explanatory texts were added to page 1: line 15-18 (highlighted in blue).
3.2. In the literature review, please use more recent works especially the ones which were published in 2020, 2021 and 2022.
Response 3.2: We thank the reviewer for the valuable comments. We have used methane pyrolysis review and hydrogen production which published in 2020, 2021 and 2022. (Journal of Energy Chemistry 58, 415-430 (2021); Applied Sciences 11, 11363 (2021); Energies 14, 3107 (2021))
3.3. Moreover, in the introduction section you can use of some other papers in the filed CO2 reforming of CH4 or dry reforming of methane (DRM) or combination of partial oxidation of CH4 and DRM. Therefore, introduction will be better. For example you can use of these papers:
- M. Sajjadi, M. Haghighi, F. Rahmani, "On the synergic effect of various anti-coke materials (Ca–K–W) and glow discharge plasma on Ni-based spinel nanocatalyst design for syngas production via hybrid CO2/O2 reforming of methane", International Journal of Energy Research, 2022, 108: p. 104810
- M. Sajjadi, M. Haghighi, J. Eshghi, "Synergic influence of potassium loading and plasma-treatment on anti-coke property of K-promoted bimetallic NiCo-NiAl2O4 nanocatalyst applied in O2-Enhanced dry reforming of CH4", International Journal of Hydrogen Energy, 2019, 44 (26): p.13397-13414
- M. Sajjadi, M. Haghighi, F. Rahmani, J. Eshghi, "Plasma-enhanced sol-gel fabrication of CoWNiAl2O4 nanocatalyst used in oxidative conversion of greenhouse CH4/CO2 gas mixture to H2/CO", Journal of CO2 Utilization, 2022, 61: p. 102037
Response 3.3: We thank the reviewer for the valuable comments. The parpers recommend by the reviewer has been cited it in the introduction (International Journal of Hydrogen Energy 44, 13397-13414 (2019); Journal of CO2 Utilization 61, 102037 (2022); International Journal of Energy Research, 108, 104810 (2022)). “Common hydrogen production methods of methane include steam reforming of methane, partial oxidation of methane, dry reforming of methane and methane pyrolysis; Dry reforming of methane can reduce CO2 emissions, but the high cost of equipment and the purification of hydrogen hinder its industrial application”. The explanatory texts were added to page 1: 31-33 and 35-37 (highlighted in blue).
3.4. Please add the specifications of the utilized column in gas chromatography? The name of that, is packed column or capillary?
Response 3.4: The column is TDX-01, Type: Φ 3 mm × 1.5 mm packed column. (The signal of CH4 can be detected at 4 min and the H2 can be detected at 0.6 min in GC-7820A)
3.5. CH4 conversion formula (in page 3, equation number 3), it seem that it is wrong. You must write in this form
(CH4 (out) – CH4 (in))/(CH4 (in))
Response 3.5: We are sorry for the mistake. We have revised the equation 3 in our manuscript.
3.6. If it is possible for you, please compare your results with the other papers. Therefore, superiority of your work will be clear.
Response 3.6: We thank the reviewer for the valuable comments. We related compare results were added in manuscript. For a comparative study, the performance of the catalyst used in this work was compared to those of the recently reported by Parkinson et al. and Nazanin Rahimi et al.. And the results clearly indicated that our work shows better methane conversion at lower temperature. The explanatory texts were added to page 7: line 241-244 (highlighted in blue). (International Journal of Hydrogen Energy 46, 6225-6238 (2021); Applied Catalysis B: Environmental 254, 659-666 (2019))
Round 2
Reviewer 1 Report
The optimization process of reaction conditions is not detailed enough, and the author needs to define the optimization process of the best preparation conditions. At the same time, the best preparation conditions provided should be able to repeat the experimental results under the operation of other researchers in other laboratories.
Author Response
We thank the reviewer for the valuable comments. We have added the optimization process of reaction conditions detail in manuscript. The Ni foam string shows better conversion rate compared with other methods. In Ni foam string experiment, the Ni foam (total:16 ; diameter of 2 cm and height of 1 cm) fixed on the gas tube. After 5 h reaction , the methane conversion could reach 19.2% at 900℃. The explanatory texts were added to page 6 : line 196-199 and line 201-203.
Reviewer 2 Report
The manuscript can be accepted in present form.
Author Response
we wish to thank the Reviewer again for the very constructive comments and suggestions to improve the quality of our manuscript. Thank you very much!
Reviewer 3 Report
Generally, it seems that Optimized process for melt pyrolysis of methane to produce hydrogen and carbon black over Ni foam/NaCl-KCl catalyst was revived in an appropriate form and also, the added answers and discussions are acceptable. In my opinion, this work is acceptable and is able to publish in the Journal of Processes.
Author Response

(The authors gave the same response as above.)
